# Machine learning of human plasma lipidomes for obesity estimation in a large population cohort

**Mathias J. Gerl**[1]*, **Christian Klose**[1], **Michal A. Surma**[1,2], **Celine Fernandez**[3],
**Olle Melander**[3,4], **Satu Männistö**[5], **Katja Borodulin**[6], **Aki S. Havulinna**[6,7],
**Veikko Salomaa**[6], **Elina Ikonen**[8], **Carlo V. Cannistraci**[9,10,11], **Kai Simons**[1,12]

**1** Lipotype GmbH, Dresden, Germany, **2** Łukasiewicz Research Network—PORT Polish Center for Technology Development, Wroclaw, Poland, **3** Department of Clinical Sciences, Lund University, Malmö, Sweden, **4** Department of Emergency and Internal Medicine, Skåne University Hospital, Malmö, Sweden, **5** Public Health Promotion Unit, National Institute for Health and Welfare, Helsinki, Finland, **6** National Institute for Health and Welfare, Helsinki, Finland, **7** Institute for Molecular Medicine Finland (FIMM-HiLife), Helsinki, Finland, **8** Department of Anatomy, Faculty of Medicine, University of Helsinki, Finland, **9** Biomedical Cybernetics Group, Biotechnology Center (BIOTEC), Center for Molecular and Cellular Bioengineering (CMCB), Department of Physics, Technische Universität Dresden, Dresden, Germany, **10** Center for Systems Biology Dresden, Dresden, Germany, **11** Complex Network Intelligence Lab, Tsinghua Laboratory of Brain and Intelligence, Tsinghua University, Beijing, China, **12** Max Planck Institute of Molecular Cell Biology and Genetics, Dresden, Germany

* gerl@lipotype.com

**Data Availability Statement:** All FINRISK data discussed in the paper can be made available to established researchers by a written application to the FINRISK Executive Board. The application

## Abstract

Obesity is associated with changes in the plasma lipids. Although simple lipid quantification is routinely used, plasma lipids are rarely investigated at the level of individual molecules. We aimed at predicting different measures of obesity based on the plasma lipidome in a large population cohort using advanced machine learning modeling. A total of 1,061 participants of the FINRISK 2012 population cohort were randomly chosen, and the levels of 183 plasma lipid species were measured in a novel mass spectrometric shotgun approach. Multiple machine intelligence models were trained to predict obesity estimates, i.e., body mass index (BMI), waist circumference (WC), waist-hip ratio (WHR), and body fat percentage (BFP), and validated in 250 randomly chosen participants of the Malmö Diet and Cancer Cardiovascular Cohort (MDC-CC). Comparison of the different models revealed that the lipidome predicted BFP the best ($R^2 = 0.73$), based on a Lasso model. In this model, the strongest positive and the strongest negative predictor were sphingomyelin molecules, which differ by only 1 double bond, implying the involvement of an unknown desaturase in obesity-related aberrations of lipid metabolism. Moreover, we used this regression to probe the clinically relevant information contained in the plasma lipidome and found that the plasma lipidome also contains information about body fat distribution, because WHR ($R^2 = 0.65$) was predicted more accurately than BMI ($R^2 = 0.47$). These modeling results required full resolution of the lipidome to lipid species level, and the predicting set of biomarkers had to be sufficiently large. The power of the lipidomics association was demonstrated by the finding that the addition of routine clinical laboratory variables, e.g., high-density lipoprotein (HDL)- or low-density lipoprotein (LDL)- cholesterol did not improve the model further. Correlation

portal is located at https://thl.fi/fi/tutkimus-ja-kehittaminen/tutkimukset-ja-hankkeet/finriski-tutkimus/tietoa-tutkijoille. More information can be obtained through finriski@thl.fi. MDC-CC data discussed in the paper will be made available to readers based on a written application to the MDC-CC steering committee (info@med.lu.se).

**Funding:** VS has been supported by the Finnish Foundation for Cardiovascular Research (http://www.sydantutkimussaatio.fi/en/foundation) and EI by the Academy of Finland (https://www.aka.fi/en, grants 307415, 312491). The funders had no role in study design, data collection and analysis, decision to publish, or preparation of the manuscript.

**Competing interests:** I have read the journal's policy and the authors of this manuscript have the following competing interests: KS is CEO of Lipotype GmbH. KS, CK and MS are shareholders of Lipotype GmbH. MJG is employee of Lipotype GmbH. VS has participated in a conference trip sponsored by Novo Nordisk and received an honorarium from the same source for participating in an advisory board meeting. He also has ongoing research collaboration with Bayer Ltd.

**Abbreviations:** BFP, body fat percentage; BIA, bioelectrical impedance analyzer; BMI, body mass index; CE, cholesteryl ester; Cer, ceramide; Chol, cholesterol; DAG, diacylglyceride; HDL, high-density lipoprotein; LC-MS, liquid chromatography-mass spectrometry; LDL, low-density lipoprotein; LPC, lysophosphatidylcholine; LPE, lysophosphatidylethanolamine; MAE, mean absolute error; MDC-CC, Malmö Diet and Cancer Cardiovascular Cohort; mol%, molar fraction; PC, phosphatidylcholine; PE, phosphatidylethanolamine; RSS, residual sum of squares; SM, sphingomyelin; TAG, triacylglyceride; WC, waist circumference; WHR, waist-hip ratio.

analyses of the individual lipid species, controlled for age and separated by sex, underscores the multiparametric and lipid species-specific nature of the correlation with the BFP. Lipidomic measurements in combination with machine intelligence modeling contain rich information about body fat amount and distribution beyond traditional clinical assays.

## Introduction

Obesity, the abnormal or excessive fat accumulation that may impair health [1], is associated with increased morbidity and mortality from diseases such as type 2 diabetes and cardiovascular disease [2, 3]. According to World Health Organization, obesity has nearly tripled since 1975, which resulted in 39% of overweight and 13% of obese adults worldwide in 2016 [1].

Obesity can be estimated in a variety of ways: Most commonly, the body mass index (BMI), a ratio of body weight-for-height [4], is used as an indicator of general adiposity. It is convenient and simple but results in varying cardiovascular and metabolic manifestations across individuals. Although BMI largely increases as adiposity increases, it does not distinguish between fat and lean mass, and therefore, individuals with greater muscle mass will also have higher BMIs [5]. The waist-hip ratio (WHR) is an easily accessible measure of body fat distribution and consists of a comparison of waist and hip circumferences. Larger WHR indicates more intra-abdominal fat and is associated with higher risk for type 2 diabetes, cardiovascular disease, and mortality [6]. Similarly, waist circumference (WC) can be used and has been considered a more straight forward and reliable measure compared with WHR [7]. Furthermore, body fat percentage (BFP) is a measure of proportion of adipose tissue in the body compared with lean mass and water [8] and is mostly determined using bioelectrical impedance in field methods. Bioelectrical impedance analysis is a repeatable, easy-to-use, and low-cost method for the estimation of BFP; however, its reliability can be influenced by various factors, including the equation used and the characteristics of the sample in which they have been validated in [9]. BFP is associated with increased all-cause mortality independently of BMI and is often suggested to be a better estimation of adiposity than BMI for prognostic and exploratory purposes [10].

The human genetic predisposition to obesity is rather low. For example, a set of 97 genetic loci have been found associated with BMI, but they accounted for only 2.7% of BMI variation [11]. Similarly, a set of 12 loci explained 0.58% of the variance in BFP [12]. Thus, the genotype may not provide sufficient information for reliable risk assessment of obesity and associated outcomes, highlighting the need for more direct, phenotypic read-outs.

Lipidomics is an omic science, which comprehensively measures the entirety of lipid molecules in a sample [13–15]—the lipid phenotype—and can be used to identify multiparametric biomarkers for disease detection, prediction, and patient stratification. For shotgun lipidomics, this can be obtained in a single mass spectrometric measurement after direct infusion of the sample. The plasma lipidome offers a plethora of information on lipids, the metabolism, and biological functions that are currently inaccessible to routine clinical lipid chemistry. This information can be used to obtain insights into many complex disease processes [16, 17]. The shotgun lipidomics technique, in which lipids are efficiently obtained from biological material by automated organic solvent extraction and measured quantitatively and reproducibly in an automated high throughput approach, allows fast screening of several thousand samples with high reproducibility [18], rendering this technology a promising tool for clinical risk assessment and precision biomedicine.

Although first lipidomic biomarkers are entering the clinic [19, 20], certain analytical standards, such as intersite reproducibility, need to be established in order to make lipidomic measurements generally accepted in clinical settings [21, 22].

Here, we applied machine learning to model obesity estimates for a lipidomics data set of the large FINRISK 2012 population cohort comprising 1,061 plasma samples [23]. We identified a complex lipidomic signature for BFP and validated the model with an independent data set of the Malmö Diet and Cancer Cardiovascular Cohort (MDC-CC) comprising randomly selected 250 plasma lipidomes [24, 25] measured on the same platform [18]. We could predict BFP with an error of 8% of its full range and explain 73% of its variation based on age, sex, and the lipidome. This lipidomic signature of obesity outperforms classical clinical lipid measures and provides fine-grained and quantitative molecular phenotype enabling stratification and identification of different obesity manifestations.

Analyzing the plasma lipidome or the metabolome [26] to estimate obesity is of course much more complicated than by direct measurement and not what we aimed for. Instead, we are investigating how the plasma lipids reflect metabolic status and whether the plasma lipidome can be used to predict health and disease. There is already ample evidence that the plasma lipidome is changing in different disease states [16, 27], and here, we show that the plasma lipidome indeed gives information beyond obesity measures and classical clinical lipid parameters, such as triglycerides and cholesterol.

We find that the lipidome gives information about the body fat distribution as measured by the WHR because a number of lipid species correlate with the WHR, even when controlled for BMI. Lipidomes show differences between the sexes, concerning lipid levels, lipid coefficients of variation, and correlations of lipid species with obesity measures. These correlation profiles were similar between the 3 obesity estimates but very different from those lipids correlating with high-density lipoprotein (HDL) cholesterol, low-density lipoprotein (LDL) cholesterol, and triglyceride levels indicating that these commonly used lipid markers only insufficiently capture molecular lipid metabolism. We discuss correlations with obesity measures and find that highest lipid impact on our modeling algorithm features 4E,14Z-sphingadiene containing sphingomyelins. Finally, we look the variation not explained by the BMI and BFP regression and find those related to other clinical parameters, such as HDL and LDL cholesterol.

## Results and discussion

We performed lipidomics analysis of 1,061 plasma samples of the FINRISK 2012 cohort (S2 Table shows clinical baseline characteristics). Plasma lipid species vary substantially between individuals and on a day-to-day basis [28, 29]. Coefficients of variation for each lipid subspecies showed population variations of 23% to 150% (S1A Fig), which is considerably larger than our 6.0% median technical coefficient of subspecies variation as assessed by reference samples (method precision). Low biological variation was found in lipid classes such as cholesterol (26%) and sphingomyelin (SM, median of 26%), whereas high variation was seen in dietary lipids like triacylglyceride (TAG) and diacylglyceride (DAG) species but also for phosphatidylethanolamine (PE) species. There are differences in variations between the sexes (S1B Fig), with TAGs varying more in males and SM varying more in females. Sex-specific differences are well documented in lipidomics studies [27, 30, 31].

### Modeling obesity

Associations of BMI and obesity with lipidomes were investigated before [27, 32], and a more detailed discussion can be found in the S1 Text. We proceeded to construct models predicting obesity from the lipidome of the FINRISK data set. Models were trained on lipid subspecies,

including age and sex (S3 Fig) as covariables. Using a Lasso model [33] trained in a cross-validation loop, we first used BMI as our obesity measure and reached a mean absolute error (MAE) of 2.5 ± 0.18 and an explained variation of 47% (S7 Table). Then, using the same procedure, we analyzed how the plasma lipidome is predicting other obesity measures compared with the models we obtained for BMI using a normalized MAE. On this comparable metric, BMI was outperformed (Fig 1A and S7 Table) by WC (MAE = 6.5 ± 0.59, 64% variation explained, WHR (MAE = 0.039 ± 0.0033, 65% variation explained), and BFP (MAE = 3.6 ± 0.33, 73% variation explained). This indicates that the lipidomic information about adiposity, as measured by WHR, WC, and BFP, is more precise than for BMI. Therefore, lipidomes contain information about the actual amount of body fat (BFP) and its distribution (WHR/WC). In the case of BFP, the high variation explained by the model is probably due to specific lipids released by the adipose tissue into the plasma. A similar notion has been reported in the case of branched-chain and aromatic amino acids [34].

We tested the presence of BFP-specific information in the lipidome by creating linear models for each lipid subspecies controlled for age and sex. This returned 141 significant lipid species after controlling for multiple testing. A similar amount of lipid species remained significant, even when the model was controlled for BMI ($n = 82$), WHR ($n = 109$), or BMI and WHR together ($n = 52$, S5 Table). A similar situation is found for WHR, for which linear models controlled for age and sex still returned similarly high amount of lipid specific for WHR ($n = 134$), when additionally controlled for BMI ($n = 103$), BFP ($n = 90$), and the combination of BMI and BFP ($n = 93$). As the relation of WHR and BFP with BMI seems nonlinear (S2 Fig), we also tested the relation using natural splines with similar results (S5 Table). All these results argue for a BFP and WHR specific but BMI independent lipid biology captured by human plasma lipidome, which is still largely unexplored.

## Different BFP models and conditions

Six different models predicting BFP were trained and their parameters learned on 796 random training samples in a cross-validation loop (Fig 1B, Results for WHR and BMI in S7 Table). Tree-based random forest [35] and stochastic gradient boosting [36] do not perform significantly better than an ordinary linear model [37] of all lipid predictors. Partial least squares [38], which is well suited for the multicollinearity characterizing lipidomic data sets, was performing better but the Lasso [33] and Cubist [39] models showed even better performance. The simple Lasso model fit the data equally well as the Cubist model, and we used it for all remaining analyses because of its simplicity and interpretability. We also tested whether normalizing absolute lipid amounts to the total lipid amount in a sample (molar fraction [mol%]) would improve the fit by removing the influence of different lipid levels between samples. However, we found no evidence of this (Fig 1C).

## Description of the BFP model

The best performing BFP Lasso model (MAE = 3.61 ± 0.33, variation explained = 73.2 ± 5%) resulted in 58 predictors, but there is also a slightly less performing Lasso model (MAE = 3.65 ± 0.33, variation explained = 72.9 ± 5.1%) with only 45 predictors within 1 standard error (S4 Fig and S6 Table). The simpler multiparametric model based on 45 predictors is essentially a subset of the complex multiparametric model based on 58 predictors (Figs 2 and S5).

The Pearson correlation network of the predictors of both Lasso models (Fig 2) shows several interesting features. Within the common lipid predictors of both BFP Lasso models, SM 34:1;2 has the greatest negative and SM 34:2;2 the greatest positive lipid $\beta$-coefficients by far

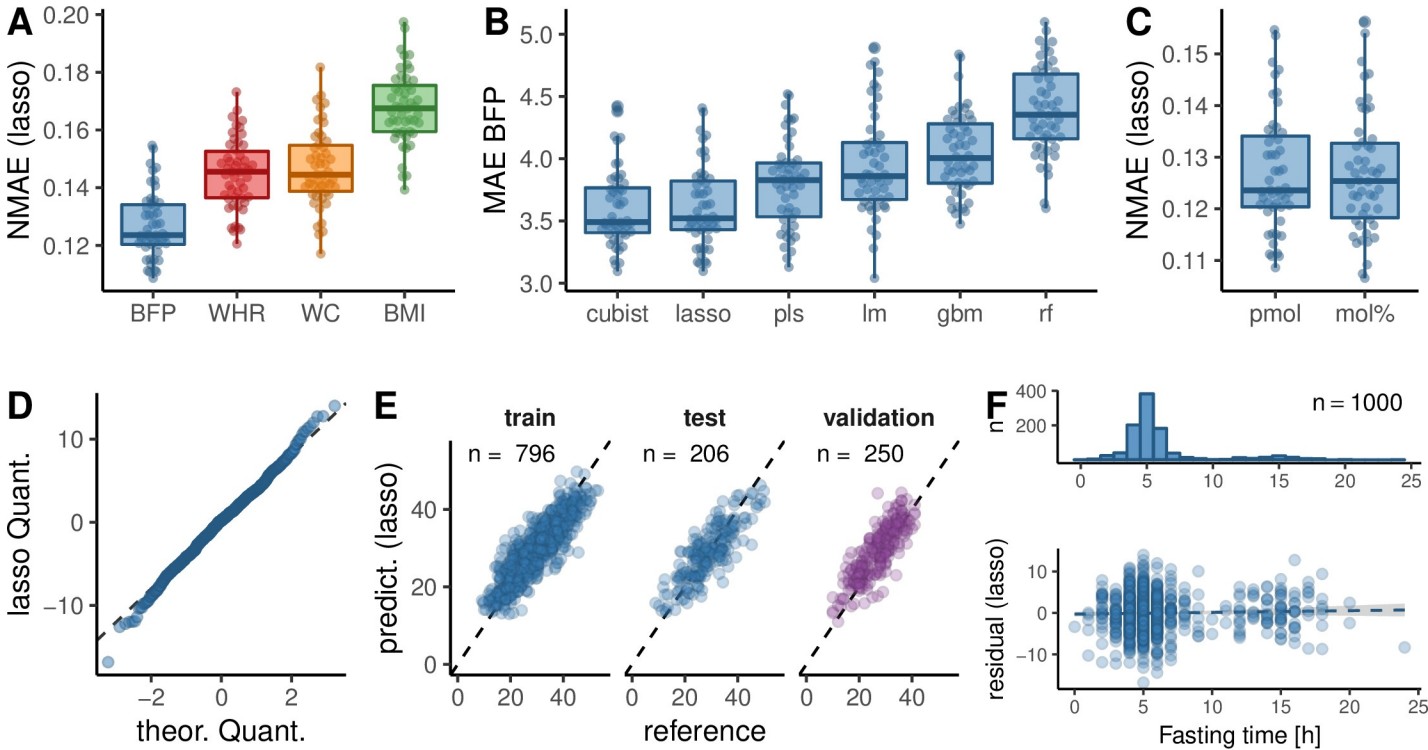

**Fig 1. Regression of obesity measures by lipidome, age, and sex.** (A) The NMAE (MAE divided by the range from the 5th to 95th percentile; S2 Fig) of different obesity measures based on Lasso regression of molar amount data. Only subjects were used, for which all obesity measures were available. (B) MAE of BFP comparing different regression algorithms on molar lipid amount data (S7 Table). (C) Lasso based NMAE of BFP comparing direct molar amounts (pmol) to molar amounts standardized to the total lipid amount within a sample (mol%). A two-sided, unpaired Mann–Whitney U test resulted in a *p*-value of 0.99. (A–C) The summary statistics of a 5× repeated 10-fold cross validation of the FINRISK training data set (80% of the data set) are shown. (D) Quantile-quantile plot of the training residuals of the Lasso BFP model against a normal distribution. (E) Original BFP values (reference) in the FINRISK training, test, and the MDC-CC validation data set plotted against the prediction of Lasso regression based FINRISK training set. *n* signifies the number of samples in each set. (F) Histogram of fasting times of subjects in the FINRISK data set and scatter plot of the Lasso residuals against fasting time, including a linear model. The slope of the linear model had a *p*-value of 0.33. BFP, body fat percentage; BMI, body mass index; gbm, stochastic gradient boosting; lm, linear model; MAE, mean absolute error; MDC-CC, Malmö Diet and Cancer Cardiovascular Cohort; mol%, molar fraction; NMAE, normalized MAE; pls, partial least squares; pmol, picomol; rf, random forest; WC, waist circumference; WHR, waist-hip ratio.

(S5 Fig and S6 Table), whereas both are correlated with each other in the correlation network within a cluster of other SM species. The additional double bond in SM 34:2;2 is likely due to an 18;2;2 long-chain base [40, 41], which is present in human plasma [41] and has been shown to be a 4E,14Z-sphingadiene [42], thus suggesting SM 18:2;2/16:0;0 as the subspecies for SM 34:2;2 in plasma [31]. SM 34:2;2 and further doubly unsaturated SMs correlate positively with BFP, i.e., SM 36:2;2 and SM 38:2;2 especially in females (S11 Table), in which they also show higher levels (S3 Fig and S4 Table, [27, 31]). The 4E,14Z-sphingadiene is suggested to be produced by an unknown desaturase, which also creates the single 14Z double bond in 1-deoxy-sphingolipids [43]. Its supposed higher activity in females results in higher levels of the respective ceramides (Cers) and SMs [27]. As SM 34:1;2 has been reported to be >96% SM 18:1;2/16:0;0 in plasma [31], it is the occurrence of 4E,14Z-sphingadiene in specifically SMs with a 16:0;0 fatty acid, which is the major correlation with BFP picked up by the Lasso models. Their significance is supported by the reduction in prediction power if the SM class is removed from the model (S8 Fig), the fact the SMs are a particularly stable lipid class in plasma (S1 Fig), and that long-chain base effects of plasma sphingolipids have been recently reported to correlate with BMI [31]. How the balance between sphingosine and 4E,14Z-sphingadiene is

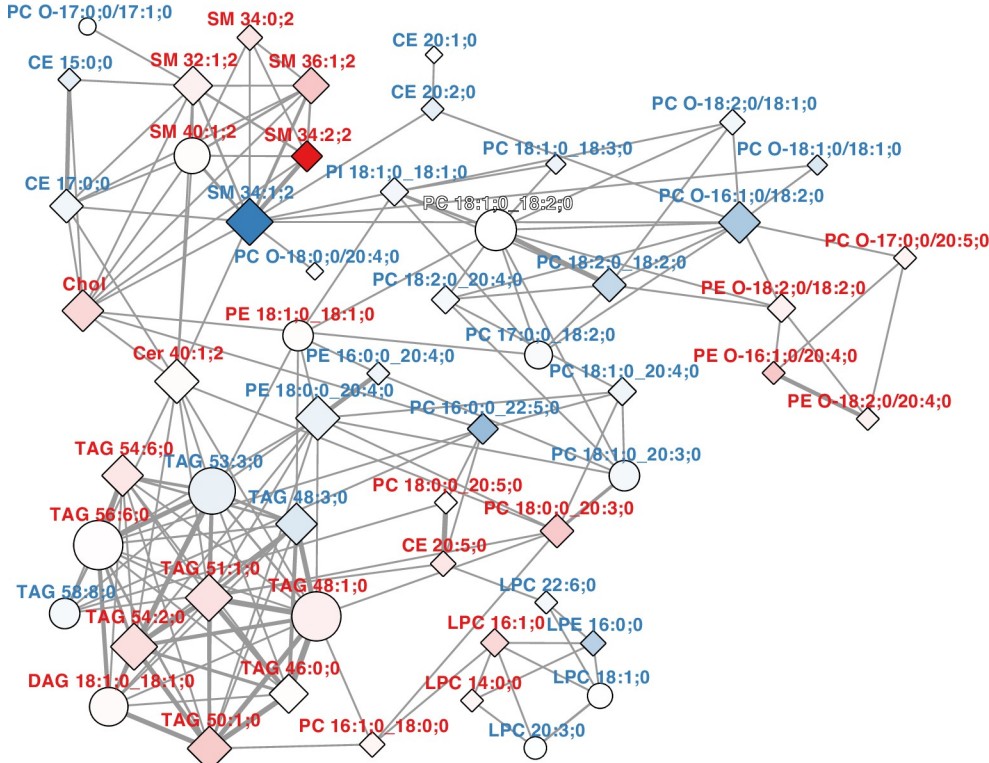

**Fig 2. Lasso model predictors.** Pearson correlation network of the lipid predictors of the best Lasso model predicting BFP with the lowest MAE and the model at 1 standard error distance (as in S4 Fig). A network cutoff of |*r*|>0.5 was used. Nodes are shaped as diamonds for predictors in both models and as circles if the predictor appears only in one model. Nodes are filled according to the β-coefficients of the model with the lowest MAE, with a gradient from blue to white for negative β-coefficients and a gradient from white to red for positive β-coefficients. Lipid labels are colored blue for negative β-coefficients and red for positive β-coefficients. Edge weights indicates the value of the correlation coefficient (*r*). All values of *r* are positive in this network. The data are reported in S6 Table, and β-coefficients are plotted in S5 Fig. BFP, body fat percentage; CE, cholesteryl ester; Chol, cholesterol; DAG, diacylglyceride; LPC, lysophosphatidylcholine; MAE, mean absolute error; PC, phosphatidylcholine; PE, phosphatidylethanolamine; PI, phosphatidylinositol; SM, sphingomyelin; TAG, triacylglyceride.

mechanistically related to the overall metabolic status and its usefulness as a general BFP bio-marker needs to be further investigated.

Associated with the SM cluster are multiple lipid predictors (cholesteryl ester [CE] 15:0;0, CE 17:0;0, and PC O-17:0;0/17:1;0) with odd chain fatty acids (Fig 2), which could be due to dairy consumption [44] or dietary fiber intake [45]. However, their association with SM and Cer species (Fig 2) might also indicate that these fatty acids are derived from hydroxylated fatty acids in glycosphingolipids or phytosphingosine [46] and therefore link the model to sphingolipids not measured in this study. Furthermore, we find a cluster of correlated lyso-lipids and of TAG species (Fig 2). TAGs with positive β-coefficients are largely consistent with common fatty markers [47]. A more detailed discussion of this observation and the association of product-to-precursor ratios of lipid metabolism enzymes to obesity measures is provided in the S1 Text.

Although the Lasso models are dominated by 2 coefficients of the sphingadiene SMs, the error of the model increases significantly, when less than 20 lipid predictors are used (S4 Fig), arguing that a single biomarker, or a small set of biomarkers, are not sufficient to predict and to faithfully capture the complex molecular scenario associated with obesity. In the FINRISK

data set, fasting duration for subjects peak around 5 hours (semifasting); however, we saw no trend in the model residuals with fasting length (Fig 1F), indicating that differences in fasting time do not have an impact on the accuracy of the prediction of BFP. This is likely due to the fact that our model is not only based on diet-derived lipids, the levels of which are acutely varying in the blood plasma, but that the predictors of the model are spread across all lipid classes except the one HexCer species (S12A Fig). For example, changes in the diet are reflected in serum TAGs within the first few hours, whereas serum CE and phospholipids reflect the last 3 to 6 weeks [48].

## Independent validation of the obesity model

Training of the BFP model on the FINRISK test set resulted in a cross-validation MAE of $3.61 \pm 0.33$ BFP units, which is about 8% of the BFP range (S2C Fig). The training error of the model was found at a MAE of 3.33 BFP units, and the mean error of the hold out FINRISK test data was at 3.84 (Fig 1E & Table 1). We validated the FINRISK based BFP model in a second, independent data set (MDC-CC), the clinical baseline characteristics of which differ from the FINRISK data set (S2 Table).

This validation resulted in a MAE of 3.67, which is only slightly above the cross-validation error obtained with the FINRISK data set. The validation also confirms that the models obtained were independent of the fasting duration, because the participants from the MDC-CC cohort were fasted over night.

The MDC-CC validation data set was measured 2 years later than the FINRISK data set on the same platform, arguing for our shotgun lipidomic approach to be highly reproducible. Taken together, these results show that we have identified a robust BFP lipidomic signature (Fig 2 and S6 Table), which was validated in an entirely independent data set. It would be interesting to see whether the model is transferable to subjects from other geographic regions with different population structures and lifestyle habits, because both data sets used originate from northern European countries.

## Comparison to a metabolomic obesity model

Recently, a metabolomic data set was used to model BMI using 49 selected metabolites. This study found that this set of metabolites explained 43% of BMI variation when age and sex were included [26]. If the model was extended to the full set of 650 metabolites measured in the study, 47% to 49% of the BMI variation could be explained. In both cases, a major fraction of the metabolites (47% and 40%, respectively) were associated with the lipid superpathway. Our

**Table 1. Reproducibility of the model.** Models were trained on the FINRISK training data set in a cross-validation loop, which results in a BFP cross validation MAE. Fitting the model on all the training data, using the best performing parameter set, results in a training MAE, testing the model on the hold out test data gives the testing error, and applying the model to the independent MDC-CC data set results in the validation error. See S7 Table for results of all models.

| MAE (BFP) | Data set | n | MAE |
|---|---|---|---|
| Cross validation | FINRISK | 796 | $3.61 \pm 0.33$ |
| Training | FINRISK | 796 | 3.33 |
| Testing | FINRISK | 206 | 3.84 |
| Validation | MDC-CC | 250 | 3.67 |

**Abbreviations:** BFP, body fat percentage; MAE, mean absolute error; MDC-CC, Malmö Diet and Cancer Cardiovascular Cohort

BMI model, although similar in many modeling aspects, is exclusively based on shotgun lipidomics. With 75 predictors in a Lasso model, it explains 47% of the BMI variation, and a model with only 50 predictors resulted in 46.5% of the variation explained. Although the population, experimental set-up, and computational modeling in the metabolomic study and in our study are not directly comparable, this suggests that the data generated with our lipidome shotgun method provide predictions of comparable quality as liquid chromatography-mass spectrometry (LC-MS) metabolomic data used in the above-mentioned study. However, the goal was achieved with a single measurement in a fully high-throughput assay. Therefore, shotgun mass spectrometry lipidomics, with its quantitative and straight-forward approach, together with fast measurements, is reproducible, robust [18], and well prepared to be used in a routine clinical setting.

Although the metabolome-derived model [26] explained only about 50% of the actual BMI variation, the metabolome-predicted BMI had improved features, such as better correlations with other clinically important variables, e.g., insulin resistance and HDL cholesterol levels. In addition, if the metabolome predicted a substantially higher BMI than the actual BMI of the subject, these subjects scored worse on a set of clinical health measures. If, however, the metabolome predicted a lower BMI than their actual BMI, the subjects scored better on the respective health measures. Because of uneven distribution of outliers in our models, we were not able to fully show the just described outlier characteristics as in Cirulli and colleagues [26]. However, when we restrict our models to a range of the FINRISK data set, in which both over- and underestimated outliers are present, we observe similar effects (S9 Fig). Still, the overpredicted outliers are in the range of low observed BMI and the underpredicted outliers in a range of high observed BMI. Therefore, the mean BMI of overpredicted samples (24.3 ± 2.0) is much lower than the mean BMI of underpredicted samples (26.9 ± 2.1). Despite this adverse setting, we could validate that individuals who had a lower BMI than predicted from the plasma lipidome had worse routine clinical laboratory values, e.g., HDL and LDL cholesterol, than those individuals whose actual BMI was higher than predicted (S9 Fig). Similar results were obtained for our BFP regression of female subjects (S10 Fig), and weaker trends were observed for male subjects (S11 Fig). Therefore, our results confirm the earlier outlier findings [26] but extend them to a lipidomic setting and also to BFP as an obesity measure. They support the conclusion that a multiparametric lipidomic estimate has a stronger predictive value on obesity than the classical predictors, improving on shortcomings in terms of total fat amount and distribution. They further show that lipidomics captures obesity-related metabolic aberrations more accurately than classical clinical parameters. The fact that outliers of the obesity predictions align with better or worse clinical laboratory values suggests overlapping markers of obesity and other diseases, e.g., a dysregulated lipid metabolism, which not only align with obesity but a range of diseases [47]. Although the lipid metabolism measured by the lipidome would be interpreted by the model as, e.g., a higher BFP, these markers might actually be hinting at other diseases. This unaccounted variation should be further explored.

## Effect of input variables and level of lipidome resolution on BFP

To test the quality of the lipidomic predictions, we compared our results with predictions of BFP based on clinical parameters (Fig 3 and S8 Table). As a zero model, we used the mean of the BFP distribution with a MAE of 7.3 (Fig 3B, −Age, −Sex, No Lipids [L]) or 0% of variation explained (S7 Fig and S9 Table). Addition of routine clinical laboratory values (e.g., total cholesterol, triglycerides, LDL cholesterol, HDL cholesterol) to the model hardly improved the BFP prediction (Fig 3B [C], MAE = 7.1 or 8.0% of variation explained). Inclusion of additional variables, e.g., smoking status or blood pressure treatment alone (Fig 3B [A], MAE = 7.2 or

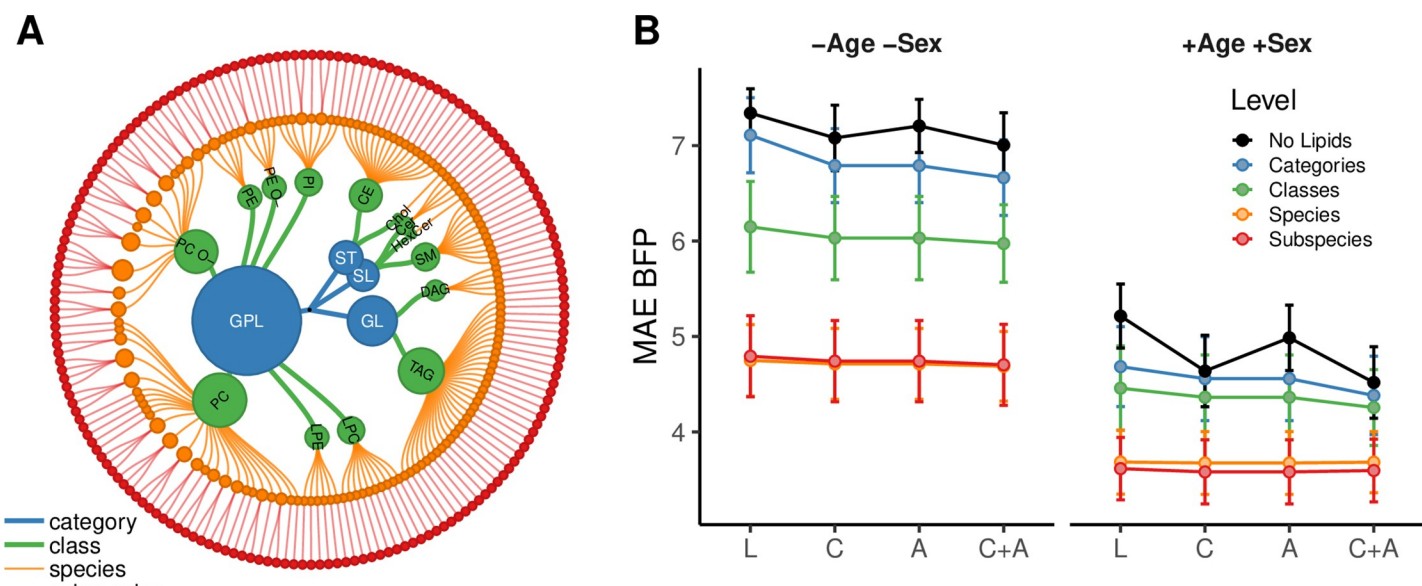

**Fig 3. Effect of different input variables and lipidome detail on the BFP regression.** (A) Lipidome hierarchy. The FINRISK lipidome, used for modeling, can be aggregated into 4 categories, 14 lipid classes, 143 species, and 183 lipid species and subspecies (S6 Fig). (B) MAE cross-validation mean and standard deviation ($n = 50$) based on lipidomes of the FINRISK training data set ($n = 796$) are shown on the y-axis. Modeling was either done without age and sex as covariables (−Age −Sex) or with (+Age +Sex). Results are colored according to lipidome detail. Either no lipid information was used (No Lipids), or lipidome information was aggregated into lipid categories (Categories), lipid classes (Classes), and lipid species (Species). Subspecies denotes the highest structural resolution possible on the platform with a mix of species and subspecies. Variables in addition to the lipidome are shown on the x-axis: No additional input [L]; routine clinical laboratory variables [C]: total cholesterol, HDL cholesterol, LDL cholesterol, triglycerides, HDL to LDL ratio, total cholesterol to HDL ratio, triglycerides to HDL ratio; additional variables [A]: blood pressure treatment, lipid treatment, smoker, pregnant, fasting, prevalent diabetes, prevalent CVD, prevalent liver disease, prevalent coronary heart disease, prevalent stroke, systolic blood pressure, diastolic blood pressure; or the combination of clinical and additional variables [C + A]. Special points are the zero model (L, No Lipids, −Age −Sex), which does not use any predictors but returns the mean of the BFP variable, and the regression only based on age and sex, (L, No Lipids, +Age +Sex), both of which are used as references for BFP predictability without regression based on L, C, or A input (S8 Table). BFP, body fat percentage; CVD, Cardiovascular disease; HDL, high-density lipoprotein; LDL, low-density lipoprotein; MAE, mean absolute error.

5.8% variation explained) or together with the routine clinical laboratory values (Fig 3B [C + A], MAE = 7.0 or 11% of variation explained) also did not improve prediction of BFP.

We then assessed how increased structural resolution of the lipidome influenced the predictive outcome (Figs 3A and S6). Already including the total molar amounts of 4 plasma lipid categories [49], glycerolipids, glycerophospholipids, sphingolipids, and sterol lipids, improved prediction outcomes to a MAE of 6.7 to 7.1 (7.0%–18% variation explained; Categories). Because this enhancement is adding to the improvement obtained by clinical parameters alone, it shows that lipid category amounts add information not contained in the other variables. The next level of structural detail is that of plasma lipid classes (e.g., PC or PE). Addition of the total molar amounts of 14 lipid classes to the BFP model further improved the prediction to 6.0 to 6.1 or 27% to 32% variation explained (Classes). The biggest improvement of the model was obtained when information of molar amounts of individual lipid molecules (143 species or a mixture of 183 species and subspecies) was used, reaching a MAE of 4.8 ± 0.43 or 55% to 57% variation explained (Species/Subspecies). Therefore, molecular lipid information is clearly superior in predicting BFP over more aggregated measures, such as HDL cholesterol, LDL cholesterol, total cholesterol, lipid categories, or lipid classes. We confirmed that the prediction based on lipid subspecies was not improved by including information on classical clinical parameters. This is expected, since LDL cholesterol, HDL cholesterol, and triglycerides have very distinct correlation patterns with the lipid subspecies profile (S16 Fig) and are, therefore, already represented in the lipidome. The correlations are in line with reported relative

amounts of lipids found in these lipoproteins [50]. We also observed multiple interesting lipid species-specific differences in these correlations, e.g., sex-specific signs for correlations of PE species with HDL cholesterol. In the case of the clinical triglycerides measurement, the major correlating lipid classes are expectedly TAG and DAG (S16 Fig). However, some highly unsaturated TAGs do not correlate strongly with the triglycerides value. Furthermore, sex-specific correlations of triglycerides are observed for cholesterol and ceramides, both of which show greater correlation coefficients in males than in females. This is in agreement with the results provided in previous studies [30, 51].

Contrary to BMI, BFP is strongly influenced by gender (S2 Fig), which is reflected by the improved prediction outcome after including age and sex variables into the model (S7 Fig). BFP predictions based on age and sex variables alone already have a MAE of 4.5 to 5.2 or 45% to 57% variation explained (Fig 3, +Age +Sex). When age and sex are considered, also routine clinical laboratory values result in improved BFP prediction (Fig 3, +Age +Sex: C, C+A). However, when the structural detail of lipid information is increased, predictions of BFP improved even further. For the model containing age, sex, and lipid subspecies, a MAE as low as 3.6 ± 0.33 or 73% variation explained was achieved. In this case, 62% of the variation not explained by age and sex is explained by the lipidome (S10 Table). These subspecies models are also not improved by the addition of clinical parameters or additional variables, which again shows that these parameters provide no additional information for BFP prediction. Similar models for BMI, WC, and WHR show comparable results (S7 Fig and S8 Table) with some variation on the magnitude of dependency on age and sex or classical clinical parameters.

We also tested whether a lipid class was necessary for prediction or could be compensated for by the other lipid classes. The most important lipid class for predicting BFP was SM. Apart from PC, the only lipid class, the removal of which reduces model performance is SM (S8 Fig). This is in agreement with complex correlation patterns observed for SM species. As mentioned above, SM 34:1;2 is inversely correlated with BFP in males, whereas SM 34:2;2 is directly correlated in females (Fig 4), both with a similar correlation estimate and the greatest positive (SM 34:1;2) and negative (SM 34:2;2) $\beta$-coefficients in the Lasso models (S5 Fig).

We conclude that the plasma lipidomes, measured by a single shotgun mass spectrometric analysis, have significantly more predictive power predicting obesity than classically used clinical parameters and that it is the resolution to molecular detail at the subspecies level that provides the relevant information. It is to be expected that similar predictive information on other metabolic states (disease and life style) are represented in the multiparametric lipidomics data.

## Lipid correlations with BFP

Although the predictive algorithm (Lasso) selects features on the basis of overall prediction error, it cannot be employed to define the individual lipidomic features correlating with BFP. Therefore, a comprehensive Spearman correlation analysis of lipid subspecies and additional features was performed for male and female subjects individually, including age as a covariable (Figs 4 and S12). The results showed that 53.5% (108) of lipid subspecies in females and 65.3% (132) in males of a total of 202 tested lipid species correlated significantly with BFP after Benjamini-Hochberg correction for multiple hypothesis testing. The BFP to lipidome correlation profile is similar to the BMI (S13 Fig), WHR (S14 Fig), and WC (S15 Fig) profiles; however, the magnitude of the correlation coefficients reflects the respective MAEs in modeling (Fig 1A), with BFP showing highest correlation coefficients and lowest error. Lipidome correlations to HDL cholesterol, LDL cholesterol, or triglycerides (S16 Fig), on the other hand, show very different profiles. As expected from the prediction results, there is a lipid subspecies-specific effect, i.e., individual lipid species showing proportional or inverse correlations, despite being

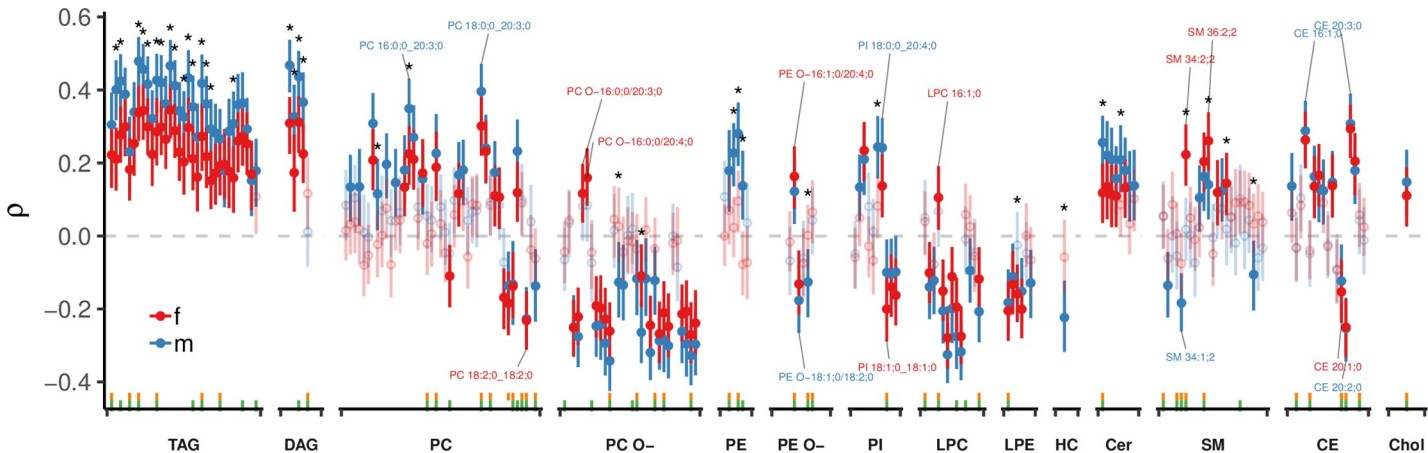

**Fig 4. Correlation of lipid subspecies with BFP.** Spearman correlation coefficients ($\rho$) and their 95% CI for each sex (male and female) and adjusted for subject age are shown for lipid subspecies. HC signifies the HexCer lipid class. Green ticks at the bottom indicate subspecies used in the model with the lowest MAE ($n = 58$), whereas yellow ticks indicate the model within 1 standard error from the lowest MAE ($n = 45$, S4 Fig). Correlations with Benjamini-Hochberg corrected $p < 0.05$ are shown with filled points, whereas correlations with $p > 0.05$ are shown with open circles and transparently. Differences between male and female correlations were compared and significant differences are indicated by an asterisk (*). A total number of 202 lipid subspecies were tested in 1,005 subjects, of those, 53.5% (108) were significantly correlated with BFP in females and 65.3% (132) in males. Some lipid subspecies of interest have been labeled. Additional correlations are shown in Fig S12 and correlation data is provided in S11 Table. BFP, body fat percentage; CI, confidence interval; MAE, mean absolute error.

in the same lipid class, e.g., CE 20:1;0 and CE 20:2;0 are inversely correlating, whereas CE 20:3;0 and 20:4;0 are proportionally correlating to BFP. All lipid classes are observed to contribute significant correlations (Fig 4 and S1 Text). These results suggest complex systemic perturbations of lipid metabolism in the obese state.

## Conclusion

We show that by exploiting the species diversity revealed by a single quantitative measurement in a lipidomics readout of a large population cohort, we can use machine learning to model and validate obesity estimates better than by using classical clinical parameters, such as total triglycerides and cholesterol. These results show that the molecular details of the plasma lipidome capture obesity-related metabolic aberrations more accurately than these classical clinical parameters. We further confirmed that outliers in the correlation between lipidomic profiles and obesity measures have clinical profiles that could predispose for later obesity-related noncommunicable diseases [26, 52]. The future challenge will be to use this technology to stratify obesity to accurately predict who will stay healthy and who will progress toward disease.

## Materials and methods

### FINRISK 2012 cohort

The National FINRISK Study is a Finnish population survey conducted every 5 years since 1972 [23]. Samples of the FINRISK 2012 underwent lipidomics measurements (1,141 randomly selected individuals) of which 1,061 were used (See S2 Table) after lipidomic quality control based on total lipid amount or disturbed lipid profile. FINRISK participants were advised to fast at least for 4 hours before the examination and avoid heavy meals earlier during the day. Measurements were obtained as described by Borodulin and colleagues [23]. BFP in the FINRISK study was measured using bioelectrical impedance device (Tanita TBF-300MA, Tanita Corporation, Tokyo, Japan).

The FINRISK 2012 survey was approved by the Coordinating Ethical Committee of the Helsinki and Uusimaa Hospital District, the participants gave a written informed consent, and the study was conducted following the principles of the declaration of Helsinki.

All data discussed in the paper can be made available to established researchers by a written application to the FINRISK Executive Board. Application portal is located at https://thl.fi/fi/tutkimus-ja-kehittaminen/tutkimukset-ja-hankkeet/finriski-tutkimus/tietoa-tutkijoille. More information can be obtained through info@med.lu.se.

### The MDC-CC

MDC-CC is a Swedish cohort designed to study the epidemiology of carotid artery disease from 1991 through 1994 [24, 25]. The MDC-CC was approved by the Regional Board of Ethics in Lund, Dnr 2009/63, and all participants provided written informed consent. A total of 250 subjects were randomly selected as a validation data set, and clinical characteristics of the study samples are presented in S2 Table. MDC-CC participant plasma samples were collected after overnight fasting. Bioelectrical impedance analyzers (BIAs) were used to estimate body composition, and BFP was calculated using an algorithm according to procedures provided by the manufacturer (BIA 103, single-frequency analyzer, JRL Systems, Detroit, IL, USA) [8].

MDC-CC data discussed in the paper will be made available to readers based on a written application to the MDC-CC steering committee (info@med.lu.se).

### Lipid nomenclature

Lipid molecules are identified as species or subspecies. Fragmentation of the lipid molecules in MSMS mode delivers subspecies information, i.e., the exact acyl chain (e.g., fatty acid) composition of the lipid molecule. MS only mode, acquiring data without fragmentation, cannot deliver this information and provides species information only. In that case, the sum of the carbon atoms and double bonds in the hydrocarbon moieties is provided. Lipid species are annotated according to their molecular composition as lipid class <sum of carbon atoms>:<sum of double bonds>;< sum of hydroxyl groups>. For example, PI 34:1;0 denotes phosphatidylinositol with a total length of its fatty acids equal to 34 carbon atoms, total number of double bonds in its fatty acids equal to 1 and 0 hydroxylations. In case of sphingolipids, SM 34:1;2 denotes a sphingomyelin species with a total of 34 carbon atoms, 1 double bond, and 2 hydroxyl groups in the ceramide backbone. Lipid subspecies annotation contains additional information on the exact identity of their acyl moieties and their *sn*-position (if available). For example, PI 18:1;0_16:0;0 denotes phosphatidylinositol with octadecenoic (18:1;0) and hexadecanoic (16:0;0) fatty acids, for which the exact position (*sn*-1 or *sn*-2) in relation to the glycerol backbone cannot be discriminated (underline "_" separating the acyl chains). On contrary, PC O-18:1;0/16:0;0 denotes an ether-phosphatidylcholine, in which an alkyl chain with 18 carbon atoms and 1 double bond (O-18:1;0) is ether-bound to *sn*-1 position of the glycerol and a hexadecanoic acid (16:0;0) is connect via an ester bond to the *sn*-2 position of the glycerol (slash "/" separating the chains signifies that the *sn*-position on the glycerol can be resolved). Lipid identifiers of the SwissLipids database [53] (http://www.swisslipids.org) are provided in S1 Table.

### Analytical process design

Samples were divided into analytical batches of 84 samples each. Each batch was accompanied by a set of 4 blank samples (150 mM ammonium bicarbonate [in water]) and a set of identical 8 control reference samples (human blood plasma). These control samples in groups of 1 blank and 2 reference samples were distributed evenly across each batch and extracted and processed together with study samples to control for background and intrarun reproducibility.

## Lipid extraction for mass spectrometry lipidomics

Mass spectrometry–based lipid analysis was performed as described by Surma and colleagues [18]. For lipid extraction, an equivalent of 1 $\mu$L of undiluted plasma was used, and plasma lipids were extracted with methyl tert-butyl ether/methanol (7:2, V:V) [54]. Internal standards (Avanti Polar Lipids, Birmingham, AL) were premixed with the organic solvents mixture. The internal standards included known amounts of: cholesterol D6 (Chol), cholesterol ester 20:0 (CE), ceramide 18:1;2/17:0 (Cer), diacylglycerol 17:0/17:0 (DAG), phosphatidylcholine 17:0/17:0 (PC), phosphatidylethanolamine 17:0/17:0 (PE), lysophosphatidylcholine 12:0 (LPC), lysophosphatidylethanolamine 17:1 (LPE), triacylglycerol 17:0/17:0/17:0 (TAG), and sphingomyelin 18:1;2/12:0 (SM). After extraction, the organic phase was transferred to an infusion plate and dried in a speed vacuum concentrator. Dried extract was resuspended in 7.5 mM ammonium acetate in chloroform/methanol/propanol (1:2:4, V:V:V). All liquid handling steps were performed using Hamilton Robotics STARlet (Hamilton Robotics, Reno, NV) with the Anti Droplet Control feature for organic solvents pipetting. Chemicals and solvents of HPLC/LC-MS analytical grade were used (Merck, Darmstadt, Germany).

## MS data acquisition

Samples were analyzed by direct infusion in a QExactive mass spectrometer (Thermo Scientific, Bremen, Germany) equipped with a TriVersa NanoMate ion source (Advion Biosciences, Ltd., Ithaca, NY). Samples were analyzed in both positive and negative ion modes with a resolution of $R_{m/z\ =\ 200} = 280,000$ for MS and $R_{m/z\ =\ 200} = 17,500$ for MSMS experiments in a single acquisition. MSMS was triggered by an inclusion list encompassing corresponding MS mass ranges scanned in 1 Da increments. Both MS and MSMS data were combined to monitor CE, DAG, and TAG ions as ammonium adducts; PC, PC O−, as acetate adducts and PE, PE O−, and PI as deprotonated anions. MS only was used to monitor LPE and LPE O− as deprotonated anions; Cer, SM, LPC, and LPC O− as acetate adducts; and cholesterol as ammonium adduct.

## Postprocessing

Spectra were analyzed with in-house developed lipid identification software based on LipidXplorer [55, 56]. TAGs are quantified as species (e.g., TAG 48:0;0). Fatty acid amounts within TAG species are achieved by distributing the total species amount by fatty acid fragment intensities. Data postprocessing and normalization were performed using an in-house developed data management system. Only lipid identifications with a signal-to-noise ratio >5, and a signal intensity 5-fold higher than in corresponding blank samples were considered for further data analysis. For FINRISK, using 3 reference samples per 96-well plate batch, lipid amounts were corrected for batch variations. An occupational threshold of 70% was applied to the data, keeping lipid species, which were present in at least 70% of the subjects. The median coefficient of subspecies variation, as accessed by reference samples, was 5.96%. In the MDC-CC data set, batch correction was applied using 8 reference samples per 96-well. Amounts were also corrected for analytical drift if the $p$-value of the slope was below 0.05 with an $R^2$ greater than 0.75 and the relative drift was above 5%. Median coefficient of subspecies variation, as accessed by reference samples, was 10.49%, and no occupational threshold was applied. For predictive modeling, lipid species were matched between the FINRISK and MDC-CC datasets.

## Data analysis

Data were analyzed with R version 3.4.2 [37] using tidyverse packages [57]. For correlations analysis, outliers were removed, which were more than 4.5 interquartile ranges from the

median of the lipid species, whereas the full set was used for predictive modeling. The data set was split for the 2 sexes, and for each subset, age-adjusted Spearman correlation coefficients ($\rho$) were calculated using the RVAideMemoire::pcor.test() function [58]. CIs were thereby created using 1,000 bootstrap resamples. When testing whether male and female correlation coefficients were significantly different from each other, the cocor.indep.groups() function from the cocor package was was used with default parameters. Correlation coefficients were significantly different if the confidence interval of the difference did not include zero [59, 60]. The correlation network was calculated with the stats::cor() function using the Pearson correlation method and pairwise complete observations. The network was visualized using Cytoscape version 3.5.0 [61].

For linear models of obesity measures with covariables, outliers were removed, which were more than 4.5 interquartile ranges from the median of the lipid species. Regression models were created by the lm() function, and 95% CIs were calculated using the confint() function. Natural splines were created with the ns() function of the splines package. Degrees of freedom of the splines were analyzed in a 10-fold cross-validation loop with MAE as readout: no effect was determined for age, and 3 degrees of freedom were used as a default for the obesity estimate covariables.

A coefficient of partial determination [62] was calculated as the proportion of variation that cannot be explained in a reduced model, using the residual sum of squares (RSS) of the full model or reduced model:

$$pR^2 = 1 - \frac{RSS_{full}}{RSS_{reduced}}.$$

## Predictive modeling

Cubist [39], Lasso [33], partial least squares [38], stochastic gradient boosting [36], random forest [35], and linear models were trained within the caret package, version 6.0–76 [63]. Input data were randomly split into a training (80%, $n = 796$) and a test data set (20%, $n = 206$) were used for all models. The input data were also filtered to contain only complete measurements of all modeled variables (BFP, BMI, WHR, WC, $n = 1,002$). Models were trained using a 5× repeated 10-fold cross-validation loop. Within the cross-validation loop data were centered and scaled (Z-score) to avoid predominance of the most abundant features, missing values were imputed by the median value of the predictor and near zero-variance variables were removed [63]. The final model was fit on all training data. MAEs are calculated by predicting the training data (training MAE), hold out test data (test MAE), and validation data (validation MAE).

## Product-to-precursor ratios

Fatty acid desaturases and elongation activities were estimated by calculating product-to-precursor ratios of sums of fatty acids in all lipids measured on the subspecies level (CE, DAG, TAG, LPC, LPE, PC, PC O−, PE, PE O−, PI) as described and discussed in work by Vessby and colleagues and Kjellqvist and colleagues [48, 64]. The following indices were used: The ratio of 16:1;0 to 16:0;0 (C16) and 18:1;0 to 18:0;0 (C18) was used to estimate the SCD1 Δ-9-desaturase (D9D) activity [48, 65, 66]. The ratio of 20:4;0 to 20:3;0 was used to estimate Δ-5-desaturase (D5D) activity [64, 66, 67]. The ratio of 18:3;0 to 18:2;0 for D6D activity [48, 66]. The ratio of 20:3;0 to 18:3;0 for ELOVL5 activity [64]. The ratio of 18:0;0 to 16:0;0 for ELOVL6 activity [68, 69]. The ratio of 16:0;0 to 18:2;0 as de novo lipogenesis index (DNL) [70].

$$SCD1/D9D\ (C16) = \frac{\sum 16:1;0}{\sum 16:0;0} \qquad SCD1/D9D\ (C18) = \frac{\sum 18:1;0}{\sum 18:0;0}$$

$$\text{D5D} = \frac{\sum 20:4; 0}{\sum 20:3; 0} \qquad \text{D6D} = \frac{\sum 18:3; 0}{\sum 18:2; 0}$$

$$\text{ELOVL5} = \frac{\sum 20:3; 0}{\sum 18:3; 0} \qquad \text{ELOVL6} = \frac{\sum 18:0; 0}{\sum 16:0; 0}$$

$$\text{Lipogenic index (DNL)} = \frac{\sum 16:0; 0}{\sum 18:2; 0}$$

## Supporting information

**S1 Text. Further discussion on lipid correlations with BFP.** BFP, body fat percentage.
(PDF)

**S1 Fig. Lipid species coefficients of variation.** (A) Coefficient of variation ($CV_i = \frac{SD_i}{\text{mean}_i}$) for all lipid subspecies, clustered and colored by lipid class. The number of measurements per species is indicated as point size ($n$). (B) Coefficient of variation for all lipid species within each sex individually are shown on the x-axis. On the y-axis Spearman correlations ($\rho$) to BFP are displayed as in Fig 4 in the main article. Coefficient of variation data is provided in the S3 Table. BFP, body fat percentage.
(PDF)

**S2 Fig. FINRISK dependent variables.** Characteristics of FINRISK variables: Distribution of (A) BMI, (B) WHR, (C) BFPs (Fat mass [%]) in the data set. (A–C) The distribution's mean, standard deviation, and number of subjects ($n$), excluding missing values, are shown in the upper left corner. Vertical dashed lines indicate the 5th and 95th percentile. Relationship of BMI to (D) WHR and (E) Fat mass according to subject sex. Lines are based on local polynomial regression fitting (loess). BFP, body fat percentage; BMI, body mass index; WHR, waist-hip ratio.
(PDF)

**S3 Fig. Sex differences of lipid species.** Volcano plot of sex differences between 571 female and 490 male subjects. *P*-values of the Mann–Whitney U test are displayed on the y-axis, fold changes of means are shown on the x-axis, which is calculated as female (f) divided by male (m) lipid levels ($FC = \frac{\text{lipid}(f)}{\text{lipid}(m)}$). Labels at the top also indicate the direction of the fold change. Points with outlines show significance <0.05 after Benjamini-Hochberg correction for multiple testing. Some of most significant SM species containing 4E,14Z-sphingadiene are labeled. All data displayed is provided in S4 Table. SM, sphingomyelin.
(PDF)

**S4 Fig. Lasso models.** MAE for BFP Lasso models with different number of predictors. The lowest MAE of 3.61 ± 0.33 was achieved by a fraction of 0.24, which used 58 lipid subspecies, whereas a model at 1 standard error distance with a MAE of 3.65 ± 0.33 used a fraction of 0.2 and 45 lipid predictors. The data are shown in S6 Table. BFP, body fat percentage; MAE, mean absolute error.
(PDF)

**S5 Fig. Lasso model predictors.** *β*-coefficients of the predictors of the best Lasso model predicting BFP with the lowest MAE and the model at 1 standard error distance (as in S4 Fig) are displayed. The data are shown in S6 Table. BFP, body fat percentage; MAE, mean absolute error.
(PDF)

**S6 Fig. Lipidome hierarchy.** The FINRISK lipidome can be aggregated into 4 categories, 14 lipid classes, 143 species, and 183 lipid species and subspecies.
(PDF)

**S7 Fig. Effect of different input variables by R$^2$.** Effect of different input variables and lipidome detail on the BFP, BMI, WC, and WHR regressions: R2 cross-validation mean and standard deviation ($n = 50$) based on the FINRISK training data set ($n = 805$) are shown on the y-axis. Zero models, which always give back the mean of the distribution, were set to 0. Modeling was done either without age and sex as covariables (−Age −Sex) or with (+Age +Sex). Results are colored according to lipidome detail. Either no lipid information was used (No Lipids), or lipidome information was aggregated into lipid categories (Categories), lipid classes (Classes), and lipid species (Species). Subspecies denotes the highest structural resolution possible on the platform with a mix of species and subspecies. Variables in addition to the lipidome input used are shown on the x-axis: No additional input [L]; routine clinical laboratory variables [C]: total cholesterol, HDL cholesterol, LDL cholesterol, triglycerides, HDL to LDL ratio, total cholesterol to HDL ratio, triglycerides to HDL ratio; additional variables [A]: Blood pressure treatment, lipid treatment, smoker, pregnant, fasting, prevalent diabetes, prevalent CVD, prevalent liver disease, prevalent coronary heart disease, prevalent stroke, systolic blood pressure, diastolic blood pressure; or the combination of clinical and additional variables [C + A]. Special points are the zero model (L, No Lipids, −Age, −Sex), which does not use any predictors but returns the mean of the respective obesity measure variable, or the regression only based on age and sex, (L, No Lipids, +Age, +Sex), both of which are used as references predictability without regression. Values are listed in S9 Table. BFP, body fat percentage; BMI, body mass index; HDL, high-density lipoprotein; LDL, low-density lipoprotein; WC, waist circumference; WHR, waist-hip ratio.
(PDF)

**S8 Fig. Dependency of the models on lipid classes.** (A) Lasso models for BFP, WHR, WC, and BMI were built on the full data set plus age and sex (indicated by −), or the lipids of the indicated class were removed (indicated by the class). R$^2$ values are shown and sorted by decreasing BFP results. (B–D) Lasso models for BFP, WHR, and BMI were built on the lipids of one lipid class only plus age and sex. R$^2$ values are shown and sorted by increasing results of the respective variable. As TAG and DAG are very correlated, both were also removed together. BFP, body fat percentage; BMI, body mass index; DAG, diacylglyceride; TAG, triacylglyceride; WC, waist circumference; WHR, waist-hip ratio.
(PDF)

**S9 Fig. Outlier analysis of BMI regression.** (A) Scatter plot of predicted BMI versus observed BMI. The dashed diagonal line represents a predicted BMI perfectly matching the observed BMI. Individual samples are shown as a scatter plot. Outliers are defined as the upper or lower 15% of the residual distribution. $n$ shows the number of FINRISK samples used for this analysis. Samples, which are not classified as outliers, are colored as Normal (BMI < 25), Overweight (25 ≥ BMI < 30), or Obese (BMI ≥ 25). Horizontal dotted lines indicate samples used for analysis. (B) Samples used for analysis: Because samples classified as "pred < obs" and "pred > obs" are not evenly distributed along the observed BMI axis; the analysis is restricted to observed BMI 21.4 to 29.73, using the lower value of the obs. BMI of the "pred < obs" and the upper value of the "pred > obs" samples as borders for the restricted area. This selects a total 699 of 1,002 (69.8%) samples, with 242 Normal, 311 Overweight, 104 "pred > obs," and 42 "pred < obs" samples. (C) Plotted variables are scaled to a common scale ranging from 0 to

1 and shown as a box plot. Outliers are plotted as black points. BMI, body mass index; obs, observed; pred, predicted.
(PDF)

**S10 Fig. Outlier analysis of BFP regression for female samples.** (A) Scatter plot of predicted BFP versus observed BFP. The dashed diagonal line represents a predicted BFP perfectly matching the observed BFP. Individual samples are shown as scatter plot. Outliers are defined as the upper or lower 15% of the residual distribution. *n* shows the number of FINRISK samples used for this analysis. Samples, which are not classified as outliers, are colored as Normal (BFP < 35) or Obese (BFP ≥ 35) [71, 72]. Horizontal dotted lines indicate samples used for analysis. (B) Samples used for analysis: Because samples classified as "pred < obs" and "pred > obs" are not evenly distributed along the observed BFP axis, the analysis is restricted to observed BFP 28 to 39, using the lower value of the obs. BFP of the "pred < obs" and the upper value of the "pred > obs" samples as approximate guides for the restricted area. The "pred > obs" value beyond obs. BFP of 40 was classified as too extreme and ignored. This selects a total 274 of 534 (51.3%) samples, with 141 Normal, 74 Overweight, 26 "pred > obs," and 33 "pred < obs" samples. (C) Plotted variables are scaled to a common scale ranging from 0 to 1 and shown as a box plot. Outliers are plotted as black points. BFP, body fat percentage; obs, observed; pred, predicted.
(PDF)

**S11 Fig. Outlier analysis of BFP regression for male samples.** (A) Scatter plot of predicted BFP versus observed BFP. The dashed diagonal line represents a predicted BFP perfectly matching the observed BFP. Individual samples are shown as scatter plot. Outliers are defined as the upper or lower 15% of the residual distribution. *n* shows the number of FINRISK samples used for this analysis. Samples, which are not classified as outliers, are colored as Normal (BFP < 25) or Obese (BFP ≥ 25) [71, 72]. Horizontal dotted lines indicate samples used for analysis. (B) Samples used for analysis: Because samples classified as "pred < obs" and "pred > obs" are not evenly distributed along the observed BFP axis, the analysis is restricted to observed BFP 19.5 to 30.8, using the lower value of the obs. BFP of the "pred < obs" and the upper value of the "pred > obs" samples as approximate guides for the restricted area. This selects a total 294 of 468 (62.8%) samples, with 134 Normal, 105 Overweight, 29 "pred > obs," and 26 "pred < obs" samples. (C) Plotted variables are scaled to a common scale ranging from 0 to 1 and shown as a box plot. Outliers are plotted as black points. BFP, body fat percentage; obs, observed; pred, predicted.
(PDF)

**S12 Fig. Correlations with BFP.** Spearman correlation coefficients ($\rho$) and their 95% CI for each sex (male and female) and adjusted for subject age are shown for (A) lipid subspecies; HC signifies the HexCer lipid class. Green ticks at the bottom indicate that this subspecies was used in the model with the lowest MAE, whereas yellow ticks indicate the model with 1 standard error from the lowest MAE (S4 Fig). A total number of 202 lipid subspecies were tested in 1,005 subjects, of those 53.5% (108) were significantly correlated with BFP in females and 65.3% (132) in males. (B) class sums; (C) fatty acid sums within the lipid classes CE, DAG, TAG, LPC, LPE, PC, PC O-, PE, PE O-, PI; (D) sums of TAG total lengths; (E) sums of TAG total double bonds; (F) sums of sphingolipid total lengths; (G) Product-to-precursor fatty acid ratios for SCD1/ *Δ*-9-desaturase (SCD-16, SCD-18), *Δ*-6-desaturase (D6D), *Δ*-5-desaturase (D5D), fatty acid elongases ELOVL5 and ELOVL6, and DNL index. Correlations with Benjamini-Hochberg corrected $p < 0.05$ are shown with filled points, whereas correlations with $p > 0.05$ are shown with open circles and transparently. Differences between male and female

correlations were compared, and significant differences are indicated by an asterisk (*). Results are provided in S11 Table. BFP, body fat percentage; CE, cholesteryl ester; DAG, diacylglyceride; LPC, lysophosphatidylcholine; LPE, lysophosphatidylethanolamine; MAE, mean absolute error; PC, phosphatidylcholine; PC O-, 1-O-alkyl- or 1-O-alkenyl-phosphatidylcholine; PE, phosphatidylethanolamine; PE O-, 1-O-alkyl- or 1-O-alkenyl-phosphatidylethanolamine; PI, phosphatidylinositol; TAG, triacylglyceride.
(PDF)

**S13 Fig. Correlations with BMI.** Spearman correlation coefficients ($\rho$) and their 95% CI for each sex (male and female) and adjusted for subject age are shown for (A) lipid subspecies; HC signifies the HexCer lipid class. A total number of 202 lipid subspecies were tested in 1,061 subjects, of those 59.9% (121) were significantly correlated with BMI in females and 60.4% (122) in males. (B) Class sums; (C) fatty acid sums within the lipid classes CE, DAG, TAG, LPC, LPE, PC, PC O-, PE, PE O-, PI; (D) sums of TAG total lengths; (E) sums of TAG total double bonds; (F) sums of sphingolipid total lengths; (G) Product-to-precursor fatty acid ratios for SCD1/$\Delta$-9-desaturase (SCD-16, SCD-18), $\Delta$-6-desaturase (D6D), $\Delta$-5-desaturase (D5D), fatty acid elongases ELOVL5 and ELOVL6, and DNL index. Correlations with Benjamini-Hochberg corrected $p < 0.05$ are shown with filled points, whereas correlations with $p > 0.05$ are shown with open circles and transparently. Differences between male and female correlations were compared and significant differences are indicated by an asterisk (*). Results are provided in S11 Table. BMI, body mass index; CE, cholesteryl ester; DAG, diacylglyceride; LPC, lysophosphatidylcholine; LPE, lysophosphatidylethanolamine; MAE, mean absolute error; PC, phosphatidylcholine; PC O-, 1-O-alkyl- or 1-O-alkenyl-phosphatidylcholine; PE, phosphatidylethanolamine; PE O-, 1-O-alkyl- or 1-O-alkenyl-phosphatidylethanolamine; PI, phosphatidylinositol; TAG, triacylglyceride.
(PDF)

**S14 Fig. Correlations with WHR.** Spearman correlation coefficients ($\rho$) and their 95% CI for each sex (male and female) and adjusted for subject age are shown for (A) lipid subspecies; HC signifies the HexCer lipid class. A total number of 202 lipid subspecies were tested in 1,051 subjects, of those 49.5% (100) were significantly correlated with WHR in females and 58.4% (118) in males. (B) Class sums; (C) fatty acid sums within the lipid classes CE, DAG, TAG, LPC, LPE, PC, PC O-, PE, PE O-, PI; (D) sums of TAG total lengths; (E) sums of TAG total double bonds; (F) sums of sphingolipid total lengths; (G) Product-to-precursor fatty acid ratios for SCD1/$\Delta$-9-desaturase (SCD-16, SCD-18), $\Delta$-6-desaturase (D6D), $\Delta$-5-desaturase (D5D), fatty acid elongases ELOVL5 and ELOVL6, and DNL index. Correlations with Benjamini-Hochberg corrected $p < 0.05$ are shown with filled points, whereas correlations with $p > 0.05$ are shown with open circles and transparently. Differences between male and female correlations were compared and significant differences are indicated by an asterisk (*). Results are provided in S11 Table. CE, cholesteryl ester; DAG, diacylglyceride; LPC, lysophosphatidylcholine; LPE, lysophosphatidylethanolamine; MAE, mean absolute error; PC, phosphatidylcholine; PC O-, 1-O-alkyl- or 1-O-alkenyl-phosphatidylcholine; PE, phosphatidylethanolamine; PE O-, 1-O-alkyl- or 1-O-alkenyl-phosphatidylethanolamine; PI, phosphatidylinositol; TAG, triacylglyceride; WHR, waist-hip ratio.
(PDF)

**S15 Fig. Correlations with WC.** Spearman correlation coefficients ($\rho$) and their 95% CI for each sex (male and female) and adjusted for subject age are shown for (A) lipid subspecies; HC signifies the HexCer lipid class. A total number of 202 lipid subspecies were tested in 1,052 subjects, of those 59.4% (120) were significantly correlated with WC in females and 57.9%

(117) in males. (B) class sums; (C) fatty acid sums within the lipid classes CE, DAG, TAG, LPC, LPE, PC, PC O-, PE, PE O-, PI; (D) sums of TAG total lengths; (E) sums of TAG total double bonds; (F) sums of sphingolipid total lengths; (G) Product-to-precursor fatty acid ratios for SCD1/$\Delta$-9-desaturase (SCD-16, SCD-18), $\Delta$-6-desaturase (D6D), $\Delta$-5-desaturase (D5D), fatty acid elongases ELOVL5 and ELOVL6, and DNL index. Correlations with Benjamini-Hochberg corrected $p < 0.05$ are shown with filled points, whereas correlations with $p > 0.05$ are shown with open circles and transparently. Differences between male and female correlations were compared and significant differences are indicated by an asterisk (*). Results are provided in S11 Table. CE, cholesteryl ester; DAG, diacylglyceride; LPC, lysophosphatidylcholine; LPE, lysophosphatidylethanolamine; MAE, mean absolute error; PC, phosphatidylcholine; PC O-, 1-O-alkyl- or 1-O-alkenyl-phosphatidylcholine; PE, phosphatidylethanolamine; PE O-, 1-O-alkyl- or 1-O-alkenyl- phosphatidylethanolamine; PI, phosphatidylinositol; TAG, triacylglyceride; WC, waist circumference.
(PDF)

**S16 Fig. Lipid species correlations with HDL, LDL, and triglycerides.** Spearman correlation coefficients ($\rho$) and their 95% CI for each sex (male and female) and adjusted for subject age are shown for lipid species and (A) HDL, (B) LDL, and (C) triglycerides, as well as lipid class sums, and (D) HDL, (E) LDL, and (F) triglycerides; correlations with Benjamini-Hochberg corrected $p < 0.05$ are shown with filled points, whereas correlations with $p > 0.05$ are shown with open circles and transparently. Differences between male and female correlations were compared, and significant differences are indicated by an asterisk (*). HDL, high-density lipoprotein; LDL, low-density lipoprotein.
(PDF)

**S1 Table. Lipid identifiers.** Identifiers of lipids used in this study to the SwissLipids database [53] (http://www.swisslipids.org) are provided.
(XLSX)

**S2 Table. Clinical baseline characteristics of the study populations.** Normal distributed variables are shown as mean with SD, and $p$-values were calculated using a ANOVA. Variables Age and total triglycerides are shown as medians with IQR, and $p$-values were calculated using a Mann–Whitney U test. IQR, interquartile range.
(XLSX)

**S3 Table. Lipid coefficient of variation.** Coefficient of variation ($CV_i = \frac{SD_i}{mean_i}$) for all lipid subspecies as shown in (S1 Fig).
(XLSX)

**S4 Table. Sex differences of lipid species.** Results of a Mann–Whitney U test of sex differences between 571 female and 490 male subjects as shown in S3 Fig.
(XLSX)

**S5 Table. Linear models for BFP and WHR with covariables.** The first sheet reports the "Number of significant lipids" for each model after correction for multiple testing. The other sheets report linear BFP or WHR models for each lipid subspecies (lipid) and covariables as indicated in the first line. For variables using natural splines variables are shown within the ns () function with the degrees of freedom indicated (e.g., df = 3). Subspecies with Benjamini-Hochberg corrected $p > 0.05$ are listed. For each model, the following information for the subspecies is provided: beta: $\beta$-coefficient; CI low/CI high: lower and higher bound of the 95% CI; $p$-value and BH: Benjamini-Hochberg corrected $p$-value. BFP, body fat percentage; WHR,

waist-hip ratio.
(XLSX)

**S6 Table. Lasso models.** Summary statistics and predictor $\beta$-coefficients are shown of the optimal model (lowest MAE) and the model within one standard error (lowest MAE + 1SE). The data are plotted in S4 Fig. MAE, mean absolute error.
(XLSX)

**S7 Table. Reproducibility of all models on all obesity estimates using MAE, NMAE, and $R^2$ as metrics.** Models were trained on the FINRISK training data set in a 5× repeated 10-fold cross-validation loop, which results in a cross-validation error. Fitting the models on all the training data, using the best performing parameter set, results in a training error, whereas testing the models on the hold out test data gives the testing error, and applying the model to the independent MDC-CC data set results in the validation error. Only subjects have been used for which all obesity measures were available ($n = 796$). MAE, mean absolute error; MDC-CC, Malmö Diet and Cancer Cardiovascular Cohort; NMAE, normalized MAE.
(XLSX)

**S8 Table. Effect of different input variables: MAE values.** Effect of different input variables and lipidome detail on the BFP, BMI, WC, and WHR regressions. Modeling was either done without age and sex as covariables (−Age, −Sex) or with (+Age, +Sex). Results are shown according to lipidome detail (Level). Either no lipid information was used (No Lipids), or lipidome information was aggregated into lipid categories (Categories), lipid classes (Classes), and lipid species (Species). Subspecies denotes the highest structural resolution possible on the platform with a mix of species and subspecies. Variables in addition to the lipidome input used are: No additional input [L], Routine clinical laboratory variables [C], Additional variables [A], or the combination of clinical and additional variables [C+A]. See caption of S7 Fig for more details. MAE cross-validation mean and standard deviation ($n = 50$) based on the FINRISK training data set, and only subjects have been used for which all obesity measures were available ($n = 796$). BFP, body fat percentage; BMI, body mass index; MAE, mean absolute error; WC, waist circumference; WHR, waist-hip ratio.
(XLSX)

**S9 Table. Effect of different input variables: $R^2$ values.** Effect of different input variables and lipidome detail on the BFP, BMI, WC, and WHR regressions. Modeling was either done without age and sex as covariables (−Age, −Sex) or with (+Age, +Sex). Results are shown according to lipidome detail (Level). Either no lipid information was used (No Lipids), or lipidome information was aggregated into lipid categories (Categories), lipid classes (Classes), and lipid species (Species). Subspecies denotes the highest structural resolution possible on the platform with a mix of species and subspecies. Variables in addition to the lipidome input used are: No additional input [L], Routine clinical laboratory variables [C], Additional variables [A], or the combination of clinical and additional variables [C+A]. See caption of S7 Fig for more details. $R^2$ cross-validation mean and standard deviation ($n = 50$) based on the FINRISK training data set, and only subjects have been used for which all obesity measures were available ($n = 796$). BFP, body fat percentage; BMI, body mass index; WC, waist circumference; WHR, waist-hip ratio.
(XLSX)

**S10 Table. Effect of different input variables: Coefficients of partial determination ($pR^2$).** Effect of different input variables and lipidome detail on the BFP, BMI, WC, and WHR regressions. Coefficients of partial determination ($pR^2$) indicate the proportion of variation that

cannot be explained in a with a "No Lipids" model (S9 Table). Modeling was either done without age and sex as covariables (−Age, −Sex) or with (+Age, +Sex). Results are shown according to lipidome detail (Level). Either no lipid information was used (No Lipids), or lipidome information was aggregated into lipid categories (Categories), lipid classes (Classes), and lipid species (Species). Subspecies denotes the highest structural resolution possible on the platform with a mix of species and subspecies. Variables in addition to the lipidome input used are: No additional input [L], Routine clinical laboratory variables [C], Additional variables [A], or the combination of clinical and additional variables [C+A]. See caption of S7 Fig for more details. pR2 cross-validation mean and standard deviation ($n = 50$) based on the FINRISK training data set, and only subjects have been used for which all obesity measures were available ($n = 796$). BFP, body fat percentage; BMI, body mass index; WC, waist circumference; WHR, waist-hip ratio. (XLSX)

**S11 Table. Correlation analyses of obesity measures, HDL, LDL, and Triglycerides.** Results of the correlation analyses as shown for BFP (S12 Fig), BMI (S13 Fig), WHR (S14 Fig), WC (S15 Fig), and HDL, LDL, and Triglycerides (S16 Fig). Values are shown for male and female separately, which are then directly compared. Estimate: Spearman correlation coefficients ($\rho$), n: number of features available, CI 95% confidence interval, BH: Benjamini-Hochberg corrected $p$-value. BFP, body fat percentage; BMI, body mass index; HDL, high-density lipoprotein; LDL, low-density lipoprotein; WC, waist circumference; WHR, waist-hip ratio. (XLSX)

## Acknowledgments

We thank Markus Damm for the creation of modeling workflows and Kai Schuhmann for helpful discussions.

## Author Contributions

**Conceptualization:** Mathias J. Gerl, Carlo V. Cannistraci, Kai Simons.

**Data curation:** Mathias J. Gerl, Christian Klose, Michal A. Surma, Katja Borodulin.

**Formal analysis:** Mathias J. Gerl, Carlo V. Cannistraci, Kai Simons.

**Funding acquisition:** Veikko Salomaa, Elina Ikonen.

**Investigation:** Christian Klose, Michal A. Surma.

**Methodology:** Christian Klose, Michal A. Surma.

**Resources:** Celine Fernandez, Olle Melander, Satu Männistö, Katja Borodulin, Aki S. Havulinna, Veikko Salomaa.

**Software:** Mathias J. Gerl.

**Supervision:** Kai Simons.

**Validation:** Christian Klose.

**Visualization:** Mathias J. Gerl.

**Writing – original draft:** Mathias J. Gerl, Carlo V. Cannistraci, Kai Simons.

**Writing – review & editing:** Christian Klose, Michal A. Surma, Celine Fernandez, Olle Melander, Satu Männistö, Katja Borodulin, Aki S. Havulinna, Veikko Salomaa, Elina Ikonen, Carlo V. Cannistraci, Kai Simons.

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
