## [Editor Report · Decision Letter 0]

3 Jun 2019

Dear Dr Gerl, 

Thank you for submitting your manuscript entitled "Machine learning of human plasma lipidomes for obesity estimation in a large population cohort" for consideration as a Research Article by PLOS Biology.

Your manuscript has now been evaluated by the PLOS Biology editorial staff as well as by an academic editor with relevant expertise and I am writing to let you know that we would like to send your submission out for external peer review.

Please re-submit your manuscript within two working days, ie. by Jun 05 2019 11:59PM.

Kind regards,

Lauren A Richardson, Ph.D

Senior Editor

PLOS Biology

---

## [Decision Letter · Decision Letter 1]

25 Jun 2019

Dear Dr Gerl,

Thank you very much for submitting your manuscript "Machine learning of human plasma lipidomes for obesity estimation in a large population cohort" for consideration as a Research Article at PLOS Biology. Your manuscript has been evaluated by the PLOS Biology editors, an Academic Editor with relevant expertise, and by several independent reviewers.

As you will read, the reviewers find many aspects of your work well done. However, they also raise some key questions that will need to be rigorously addressed in a revision. Of particular note, two of the reviewers question how useful this model is and how it will benefit other clinicians and researchers. We will need to be convinced by your justification to pursue this manuscript further for publication. 

In light of the reviews (below), we will not be able to accept the current version of the manuscript, but we would welcome resubmission of a much-revised version that takes into account the reviewers' comments. We cannot make any decision about publication until we have seen the revised manuscript and your response to the reviewers' comments. Your revised manuscript is also likely to be sent for further evaluation by the reviewers.

Your revisions should address the specific points made by each reviewer. Please submit a file detailing your responses to the editorial requests and a point-by-point response to all of the reviewers' comments that indicates the changes you have made to the manuscript. In addition to a clean copy of the manuscript, please upload a 'track-changes' version of your manuscript that specifies the edits made. This should be uploaded as a "Related" file type. You should also cite any additional relevant literature that has been published since the original submission and mention any additional citations in your response. 

Before you revise your manuscript, please review the following PLOS policy and formatting requirements checklist PDF: http://journals.plos.org/plosbiology/s/file?id=9411/plos-biology-formatting-checklist.pdf. It is helpful if you format your revision according to our requirements - should your paper subsequently be accepted, this will save time at the acceptance stage.

Please note that as a condition of publication PLOS' data policy (http://journals.plos.org/plosbiology/s/data-availability) requires that you make available all data used to draw the conclusions arrived at in your manuscript. If you have not already done so, you must include any data used in your manuscript either in appropriate repositories, within the body of the manuscript, or as supporting information (N.B. this includes any numerical values that were used to generate graphs, histograms etc.). For an example see here: http://www.plosbiology.org/article/info%3Adoi%2F10.1371%2Fjournal.pbio.1001908#s5.

Upon resubmission, the editors will assess your revision and if the editors and Academic Editor feel that the revised manuscript remains appropriate for the journal, we will send the manuscript for re-review. We aim to consult the same Academic Editor and reviewers for revised manuscripts but may consult others if needed.

We expect to receive your revised manuscript within two months. Please email us (plosbiology@plos.org) to discuss this if you have any questions or concerns, or would like to request an extension. At this stage, your manuscript remains formally under active consideration at our journal; please notify us by email if you do not wish to submit a revision and instead wish to pursue publication elsewhere, so that we may end consideration of the manuscript at PLOS Biology.

When you are ready to submit a revised version of your manuscript, please go to https://www.editorialmanager.com/pbiology/ and log in as an Author. Click the link labelled 'Submissions Needing Revision' where you will find your submission record. 

Sincerely,

Lauren A Richardson, Ph.D

Senior Editor

PLOS Biology

Reviews

Reviewer #1: Jens Nielsen, signed review

This is a very interesting paper describing the use of shut-gun lipidomics for profiling of obese subjects. Lipidomics data are used together with traditional measurements such as BMI, WC and WHR to identify a subset of lipid measurements that can be used to predict phenotypes. The authors use non-linear regression analysis (machine learning) for this analysis, and the derived model is shown to have good predictive strength. I think there are particular two interesting findings from the study: 1) that a small set of biomarkers are not sufficient to predict and capture the complex phenotypes in obese subjects, and 2) through a validation cohort the it is found that the exact timing of fasting is not influencing the set of biomarkers.

I only have one major comment and one minor comment.

Major comment:

There is no discussion of the possible molecular mechanisms associated with the biomarkers. The authors could look into whether these are mainly diet associated or are they associated with certain features, e.g. inflammation. I know it will be hard to have a detailed mechanistic discussion/explanation, but some insight along this line would significantly enrich the paper.

Minor comment:

It is stated that the 45 lipid species in the reduced model are essentially the same as the 58 lipid species in the better model. I suggest the authors quantify this statement instead of just saying essentially. Why not give the exact overlap in the lipid species in the two models. This could also be used in the discussion I am requesting on mechanisms, as overlapping lipid species are likely associated with some sort of mechanisms.

---------------

Reviewer #2: 

The authors used two sets of lipidomics data ("1061 participants of the FINRISK 2012 population cohort" and "250 randomly chosen participants of the Malmö Diet and Cancer Cardiovascular Cohort") to build and test the regression model for predicting body mass index (BMI), waist circumference (WC), waist-hip ratio (WHR) and body fat percentage (BFP) with consideration of gender and age. In the conclusion, the authors conclude that “we can use machine learning to model and validate obesity estimates better than by using classical clinical parameters and find lipid specific differences between the individual estimates.” My major concern is that as the conventional standard of obesity such as the "body mass index (BMI), waist circumference (WC), waist-hip ratio (WHR) and body fat percentage (BFP)" are easier to obtain and measure, whether the authors can demonstrate additional benefits of this model besides finding small lipid molecules associated with the obesity indicators.

Other comments:

1: Page 7 “Five different models predicting BFP were trained and their parameters learned on 796 random training samples in a cross-validation loop (Fig. 1B, Results for WHR and BMI in Table S2).” There are 6 methods in Figure 1B, not 5.

2: Figure 1C. When comparing the two groups, please perform statistical test(s).

3: Figure 2B. As age and gender have strong weight in the model is very obvious, you can build a model of age and gender separately to see how important it is.

4: Figure S3. The meaning of the X axis is not clear, which side represents Male/Female?

5: Figure S8B, S9B, S10B. How to choose the restricted area? What is the percentage of samples selected as a whole sample? And the restricted area is close to the middle, but a large number of outliers are not in this range.

6: Table S2. Please add the test set and validation set results.

---------------

Reviewer #3: 

Gerl et al employed a novel mass spectrometric shotgun approach and measured the levels of 183 plasma lipid species in participants of FINRISK 2012 population cohort comprising 1061 plasma samples. Based on this data, authors performed advanced machine learning using different predictive models and identified the association of lipid profile and information about body fat amount and distribution. The conclusion was further validated using an independent dataset of the Malmo Diet and Cancer Cardiovascular Cohort comprising randomly selected 250 plasma lipidomes. It is an interesting study but there are some concerns.

1. Authors need to better explain the merit of this study. Are lipid predictors predict development of obesity at the moment of in future? The study would be of great value if lipid predictors were measured before individuals developed obesity yet, but it would be of little value if lipid predictors simply separate obese and non-obese individuals at the moment, which wouldn’t need complex lipid measurement and modelling. 

2. Line 109, “Each batch was accompanied by a set of 4 blank samples (150 mM ammonium bicarbonate (in water)”. It is not clear why 150 mM ammonium bicarbonate is chosen as blank samples instead of reconstitution solvent “7.5 mM ammonium acetate in chloroform/methanol/propanol”.

3. In method section, line 118 the authors described “Internal standards were pre-mixed with the organic solvent mixture. “ More details are needed to assess the robustness of this method. For instance, are internal standards pre-mixed with the organic solvent mixture right before each batch? Organic solvent used for lipid extraction is very volatile so it is challenging to have internal standards in such solvent for long term with consistent concentrations. In addition, what is the volume of organic solvent mixture used in this study? It is generally challenging to consistently transfer small amount of organic solvent due to the less retention on pipette tips. 

4. In MS data acquisition section, “both MS and MSMS data were combined to 136 monitor CE, DAG and TAG ions as ammonium adducts”. The authors bypassed LC chromatography. It is unclear how authors dealt with in-source fragmentation issues. For instance, TAG would contribute to DAG signals through in-source fragmentation. PC would contribute to lyso PC signals due to in-source fragmentation. Without LC separation, it is hard to tell how authors distinguish these species. 

5. Was relative intensity or absolute concentration of lipids used for modelling?

---

## [Editor Report · Decision Letter 2]

30 Jul 2019

Dear Dr Gerl,

Thank you for submitting your revised Research Article entitled "Machine learning of human plasma lipidomes for obesity estimation in a large population cohort" for publication in PLOS Biology. 

The Academic Editor and I have now assessed your revised manuscript and we're delighted to let you know that we're now editorially satisfied with your manuscript. We will publish your study, assuming you are willing it modify it to meet our production requirements. Congratulations!

Before we can formally accept your paper and consider it "in press", we need to ensure that your article conforms to our guidelines. A member of our team will be in touch shortly with a set of requests. As we can't proceed until these requirements are met, your swift response will help prevent delays to publication.

Please note that you may have the opportunity to make the peer review history publicly available. The record will include editor decision letters (with reviews) and your responses to reviewer comments. If eligible, we will contact you to opt in or out.

Sincerely,

Lauren A Richardson, Ph.D 

Senior Editor

PLOS Biology

ETHICS STATEMENT:

The Ethics Statements in the submission form and Methods section of your manuscript should match verbatim. Please ensure that any changes are made to both versions.

-- Please provide the information for the approval of both the FINRISK and MDC-CC studies, including information about the form of consent (written/oral) given for research involving human participants. All research involving human participants must have been approved by the authors' Institutional Review Board (IRB) or an equivalent committee, and all clinical investigation must have been conducted according to the principles expressed in the Declaration of Helsinki.

DATA POLICY:

Please ensure that your Data Statement in the submission system accurately describes where your data can be found. For the MDC-CC application information, please provide an institutional email address (rather than a single person) so that the data can still be accessed even if that person departs the organization.

---

## [Editor Report · Decision Letter 3]

4 Sep 2019

Dear Dr Gerl,

On behalf of my colleagues and the Academic Editor, Jason W. Locasale, I am pleased to inform you that we will be delighted to publish your Research Article in PLOS Biology. 

Early Version

PRESS 

Kind regards,

Sofia Vickers

Senior Publications Assistant

PLOS Biology

On behalf of, 

Lauren Richardson,

Senior Editor

PLOS Biology